# INPUT SPACE MODE CONNECTIVITY IN DEEP NEURAL NETWORKS

**Jakub Vrabel** [*†]
CEITEC, Brno University of Technology

**Ori Shem-Ur**
Tel Aviv University

**Yaron Oz**
Tel Aviv University

**David Krueger**
Mila, University of Montreal

## ABSTRACT

We extend the concept of loss landscape mode connectivity to the input space of deep neural networks. Mode connectivity was originally studied within parameter space, where it describes the existence of low-loss paths between different solutions (loss minimizers) obtained through gradient descent. We present theoretical and empirical evidence of its presence in the input space of deep networks, thereby highlighting the broader nature of the phenomenon. We observe that different input images with similar predictions are generally connected, and for trained models, the path tends to be simple, with only a small deviation from being a linear path. Our methodology utilizes real, interpolated, and synthetic inputs created using the input optimization technique for feature visualization. We conjecture that input space mode connectivity in high-dimensional spaces is a geometric effect that takes place even in untrained models and can be explained through percolation theory. We exploit mode connectivity to obtain new insights about adversarial examples and demonstrate its potential for adversarial detection. Additionally, we discuss applications for the interpretability of deep networks.

## 1 INTRODUCTION

The high-dimensional nature of the parameter space in deep neural networks (DNNs) gives rise to the intriguing phenomenon of loss landscape mode connectivity. This concept reveals that different solutions of the model, achievable through stochastic gradient descent (SGD) training, are not isolated but rather interconnected by simple paths of low loss (Garipov et al., 2018; Draxler et al., 2018). This observation challenges the classical view of the loss landscape, which is traditionally perceived as containing many equivalent, discrete minima (Kawaguchi, 2016).

Connections between the modes can be as simple as linear in specific cases (Nagarajan & Kolter, 2019), while they are often quadratic or piece-wise linear in the majority of cases. Linear mode connectivity (LMC) can be associated with the consistency of mechanisms employed by models for predictions (Lubana et al., 2023) (e.g., background vs. object in image classification models). Interestingly, LMC relates to the concept of lottery tickets and their stability to SGD noise after a few initial training iterations (Frankle et al., 2020).

Mode connectivity can be utilized for enhanced ensembling strategies to create more diverse ensembles (Garipov et al., 2018). This approach can significantly boost the accuracy and generalization capabilities of models, especially in handling out-of-distribution data (Fort et al., 2020). Furthermore, parameter space mode connectivity can be employed to repair backdoored or error-injected models (Zhao et al., 2020).

The exact causes for mode connectivity within neural networks remain elusive. Architectural symmetries (node permutation) are known contributors (Zhao et al., 2023) but do not explain the full picture. Commonly used loss functions also significantly contribute to the degeneracy of the pa-

---

[*]Correspondence to: `jakub.vrabel@ceitec.vutbr.cz`.
[†]Work partially performed during a research visit at the University of Cambridge.

rameter space. Our work further explores the hypothesis that mode connectivity is a more general phenomenon of high dimensional geometry, partially studied in Fort & Jastrzebski (2019).

As we demonstrate, the concept of mode connectivity can be generalized to the input space of a neural network. In contrast to the originally studied parameter space, where data are fixed and model (parameters) variable, we employ an inverted setup (fixed model, variable data). We utilize real, virtual (interpolated), and synthetic inputs, which are obtained by input optimization (Olah et al., 2017). Considering the cross-entropy loss, we prepare and compute several examples of *optimal inputs* (natural images, high-frequency patterns, and adversarial examples) and show that they are mode connected in vision models for classification. Insights about the connectivity reveal the potential for adversarial detection and interpretability of DNNs. We propose that the existence of input space mode connectivity can be explained through high-dimensional percolation, which is a purely geometrical phenomenon. We also note that trained networks tend to contain simple paths with only small deviations from linear paths for many real input pairs, and we speculate that this could be an aspect of the implicit regularization of deep learning.

The main motivations for studying input space mode connectivity are: 1) advancing the scientific understanding of deep learning, i.e., bridging the gap between theory and practice; 2) providing new insights into interpretability by understanding the nature of adversarial examples and potential for exploring the class-optimal data manifold; and 3) enhancing adversarial robustness through adversarial detection.

**Contributions**

- We extend the concept of mode connectivity from the original parameter space to the input space of (vision) DNNs.
- We observe fundamental differences in connectivity between loss-minimizing inputs across various setups (real vs. real images, real vs. adversarial attacks, synthetic images) that allow us to distinguish between them.
- We propose a novel method for detecting adversarial examples that outperforms baselines for advanced attacks (deepfool (Moosavi-Dezfooli et al., 2016) and Carlini-Wagner (C&W) (Carlini & Wagner, 2017)).
- We conjecture that arbitrary inputs with the same prediction are mode connected in randomly initialized neural networks. We provide a proof sketch of this conjecture, based on percolation theory (Duminil-Copin, 2017), along with empirical evidence.

## 2 RELATED WORK

Prior work of Jacobsen et al. (2019) studied the phenomenon of excessive input invariance in neural networks. This research later inspired Balasubramanian et al. (2023) to explore a specific form of input space connectivity between real inputs and so-called invariance-based adversarial examples (referred to as *blind spots*). Remarkably, these blind spots can closely resemble images from different classes with entirely different semantics than the reference image, yet they maintain the same model prediction confidence along the linear interpolant paths to the reference. In contrast, we study the connectivity between almost any pair of inputs that have similar model outputs. The major difference from (Balasubramanian et al., 2023) is that invariance-based adversarial examples are (from the model's perspective) out-of-distribution pathological samples, whereas we consider all possible inputs. Moreover, we empirically and theoretically demonstrate that (loss-based) input space mode connectivity exists in general, including untrained networks.

Interpolating pairs of real input samples of different classes was proposed as an advanced data augmentation procedure called *mixup* (Zhang et al., 2018)). Mixup was shown to improve generalization and adversarial robustness, but mode connectivity was not discussed.

Our work is directly motivated and inspired by seminal research on parameter space mode connectivity (Garipov et al., 2018; Draxler et al., 2018). We leverage various established methods and analytical tools to explore connectivity in the input space. Notably, we adapted a simplified algorithm to bypass loss barriers from Fort & Jastrzebski (2019). In the same work, the authors proposed a phenomenological model of the high-dimensional loss landscape that allows analysis of the training dynamics and mode connectivity. Yunis et al. (2022) identified a high-dimensional convex hull of low loss between multiple loss minimizers, which is relevant to the discussion about the optimal

input manifold. Simsek et al. (2021) show that even originally discrete minima can be connected by adding a single neuron to each layer, highlighting the significance of space dimensionality and generality of the phenomenon.

We also build on the feature visualization by optimization technique (Olah et al., 2017) to create optimal inputs and to prepare adversarial attacks. For the discussion about adversarial detection, we follow and adapt the benchmark from Harder et al. (2021).

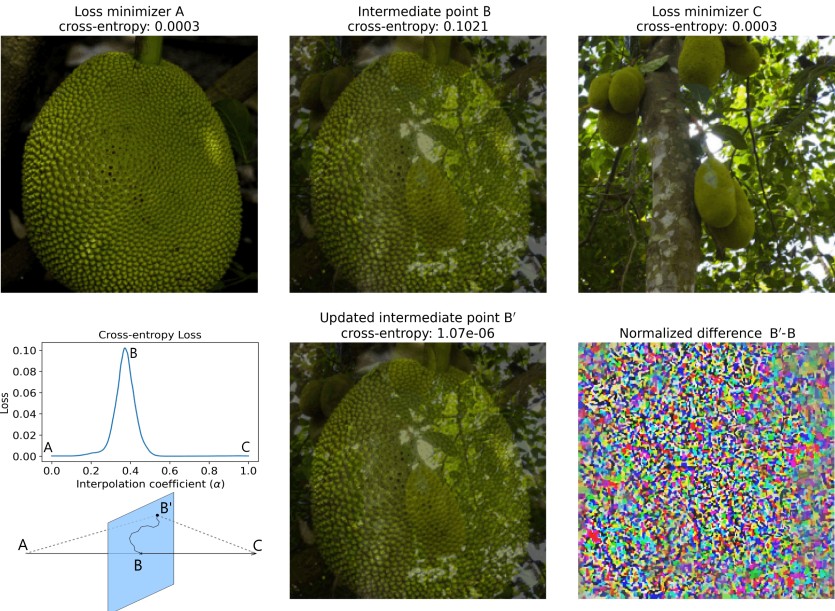

Figure 1: Barrier between two input modes. Two examples (A and C) that minimize loss for the class *jackfruit* are shown. The interpolated point B exhibits maximal loss. **Left bottom:** Cross-entropy loss for interpolated virtual points between A and C; a diagram illustrating the interpolation paths and subsequent optimization of point B, constrained within the orthogonal hyperplane to vector AC. **Middle bottom:** The optimized point B' minimizes the loss and is nearly indistinguishable from the original point B. **Right bottom:** Normalized difference pattern B'−B, where the maximum pixel value of the difference pattern is 1–5% of the maximum pixel in the original image.

## 3  METHODOLOGY

### 3.1  MODE CONNECTIVITY (FROM PARAMETER TO INPUT SPACE)

We define a parameter space mode as a set of parameters $\theta_i$ that minimizes the loss $\mathcal{L}$ (e.g., cross-entropy for classification task) for a given dataset. That is

$$\theta_i = \underset{\theta}{\arg\min}\ \mathcal{L}(f(\mathcal{D};\theta_i)), \tag{1}$$

where $f(\cdot;\theta)$ is a neural network output, and $\mathcal{D}$ is a dataset that contains both data samples $\mathcal{X}$ and ground truth $y$. Loss minimizers in the parameter space are usually obtained through SGD, with diversity arising simply from using different initializations and/or exploiting other sources of randomness such as the sequence in which examples are presented to the model during SGD-based training (i.e., *SGD noise*).

Analogically, the input space mode for a selected class $y_i$ is a single data example

$$x_i = \underset{x}{\arg\min}\ \mathcal{L}(f(x_i;\theta), y_i), \tag{2}$$

where in contrast to the parameter space, the model is fixed (pretrained). Input modes could be either real data examples with loss below a certain small threshold $\delta$ or obtained through an optimization process (see Section 3.2).

After observing two (or potentially more) modes $x_A$ and $x_C$, we linearly interpolate them to create virtual points $x_{A \to C}(\alpha)$ as

$$x(\alpha) = \alpha x_i + (1 - \alpha)x_j, \tag{3}$$

where $\alpha \in [0, 1]$ is uniformly sampled. In general, a loss barrier exists between the two modes (Fig. 1 left bottom). The barrier can be bypassed via optimization of the highest-loss point $x_B$, which results in modes connected by simple (partially linear) paths of low loss (as described in Section 4.1). The optimization follows the procedure described in Section 3.2. In the following sections, we use a simplified notation $x_A \equiv A$ for points/images, and A→C denotes a path between A and C.

## 3.2 Feature visualization by optimization (FVO)

FVO is a technique for post-hoc interpretability of DNNs (Olah et al., 2017). It starts from a small noise $x_i$ at the input of a network that is forward propagated to the layer of interest. We aim to find inputs that activate a selected neuron (or neurons) within the layer of interest. Neurons can be either maximized or activated to a specific pattern through a loss function. The input is iteratively updated w.r.t. the gradient of the loss, similarly to the standard gradient descent training. Note that this optimization process does not produce an unique result, and so multiple distinct examples can be generated and their (input space) mode connectivity studied (see App. A).

The optimization procedure can be easily adapted for real images. We start from the maximal loss intermediate point B and optimize until we reach B', which minimizes the loss (see left bottom diagram in Figure 1). The optimization is constrained within the orthogonal hyperplane to the vector AC, representing a simplified method for connecting modes inspired by Fort & Jastrzebski (2019).

## 4 Experiments

In this section, we provide empirical evidence for the input space mode connectivity in pretrained vision models and discuss potential applications of the phenomenon. First, we show how the two modes can be connected through the optimization of the loss barrier. Second, we study and exploit the connectivity between adversarial attacks and natural images to create an algorithm for adversarial detection. Third, we show that the connectivity exists for untrained (randomly initialized) networks, which validates our theoretical result in Section 5. Last, (in Appendix A) we discuss the potential for interpretability of DNNs.

We used cross-entropy loss for all experiments unless explicitly stated otherwise. Experiments have minimal compute requirements and were carried out on a regular PC (equipped with GTX 1650 4GB) or cloud services (T4 16GB). Sections 4.1, 4.2, and Appendix A use GoogLeNet(ImageNet); subsection 4.2.1 uses VGG-16(CIFAR10/100); section 4.3 uses ResNet18.

## 4.1 Input space connectivity

Vision models trained for classification tasks provide an ideal setup to observe input space mode connectivity. To demonstrate this phenomenon in a realistic and practically relevant setting, we used GoogLeNet (Inception v1) (Szegedy et al., 2014) that was pretrained on the ImageNet (Russakovsky et al., 2015) and is available from the PyTorch torchvision library (Paszke et al., 2019). First, we selected representative low-loss examples from the validation dataset, which have cross-entropy loss in the order of 1e-4 or lower. These examples were linearly interpolated with 1,000 uniformly distributed points. After identifying the interpolated point of maximum loss B, we started a constrained optimization process, encompassing 1024 iterations, which resulted in a good loss minimizer B' (Fig. 1). The Adam optimizer (Kingma & Ba, 2017) was used with a learning rate of 0.005. We added regularization to the optimization with two additional terms: deviation penalty $P_{\mathrm{MSE}}$ between B and B', and a high-frequency term $P_{\mathrm{hf}}$ that penalizes changes in adjacent pixels (see definitions in App. B). The weights for these terms, $\lambda_{\mathrm{MSE}} = 0.1$ for the deviation, and $\lambda_{\mathrm{hf}}$ ranging from $1e - 8$ to $5e - 6$ for the high-frequency term, were determined heuristically, varying by class and setup.

A new path (A→B'→C) was created by merging the linear interpolations between A and B', and between B' and C. We sampled 500 intermediate points equally from both segments of the new path, disregarding the fact that the barrier typically does not lie exactly at the midpoint of A→C. The A→B'→C path exhibits low loss while connecting two input modes (Fig. 2). Although small *secondary* barriers are still present in the A→B'→C path, we neglect them as far as they are below a certain threshold ($\delta = 0.001$ considered here). By repeating the optimization process on new segments independently, we can easily bypass secondary barriers. More examples and qualitatively consistent results from Vision Transformer (trained on ImageNet), and ResNet18 (He et al., 2015), MLP, and CNN (trained on CIFAR-10 (Krizhevsky, 2009)) are shown in the Appendix G. Images

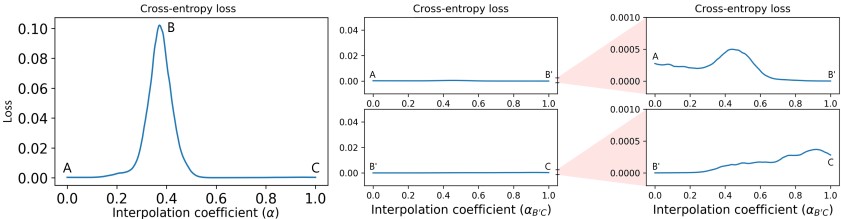

Figure 2: Bypassing the barrier. After optimizing the intermediate point B, we obtained the updated point B', which linearly connects to both A and C. Note that small *secondary* barriers, below the threshold of $\delta = 0.001$ loss, are disregarded.

along the two studied paths closely resemble each other in corresponding pairs, as shown in Figure 3. In fact, their difference is the identical pattern depicted in the right bottom of Figure 1, scaled in intensity proportionally to the interpolation factor $\alpha$. This is likely due to the process we use to find B', which resembles the way gradient descent is used to find adversarial examples, the differences being that we restrict the search to a hyperplane and do require B' to remain close to B.

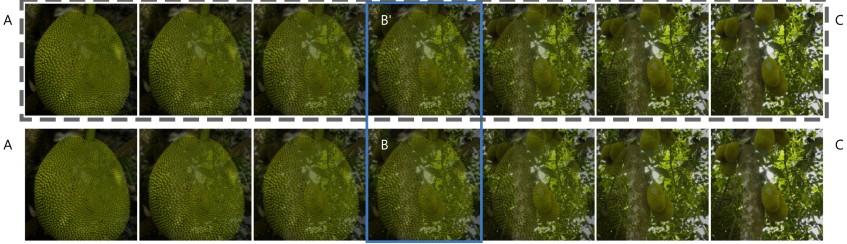

Figure 3: Images along two paths. We sampled and compared images along the updated path A→B'→C (dashed rectangle) and the original linear path A→C (bottom row). These paths are illustrated in the diagram in Figure 1. The high-loss intermediate point B and its optimized version B' are denoted by the blue vertical rectangle.

## 4.2 ADVERSARIAL ATTACKS

In this section, we demonstrate a fundamental difference in the connectivity between real–real and real–adversarial image pairs. We subsequently leverage this insight for adversarial detection.

Adversarial examples in vision models are images that are generated by perturbing a *source* image in a way that either minimizes the loss for a selected target class (in targeted attacks) or maximizes the loss for the true class (in untargeted attacks), causing the model to misclassify the image. The source image and the optimized adversarial example are often almost indistinguishable to a human observer. The process of generating and employing adversarial images to deceive the model is known as an adversarial attack (see Fig. 4).

We find that adversarial examples, even when significantly different in terms of human visual perception, are mode-connected to "true" real images (correct class minimizers). Using ImageNet and GoogLeNet, we picked a low loss example from a selected class (here *golf ball*) as the mode A.

Then, a source image K from a different class (*revolver*) was optimized to obtain an adversarial example K' (K' minimizes the same class as the A). We used Adam (learning rate 0.005), 1024 iterations, cross-entropy loss with penalizations ($\lambda_{\mathrm{MSE}} = 0.1$ for image deviation term $P_{\mathrm{MSE}}$ and $\lambda_{\mathrm{hf}} = 1e - 7$ for high-freq. penalty $P_{\mathrm{HF}}$, see definitions in App. B). From here, the mode connecting procedure is identical to the previously described: 1) Detect the maximal loss point B, 2) Optimize B to get B', 3) Interpolate over the new path A→B'→K'. However, after a single round of this iterative procedure, non-negligible secondary barriers emerge in paths A→B' and B'→K', which exceed the loss threshold $\delta$ by a considerable margin (as depicted in Fig. 5). Secondary barriers can be successfully bypassed by a repeated application of the procedure to both segments of the A→B'→K' path individually. As a result, the final low-loss path will necessarily be more complex than for real–real modes addressed in Section 4.1. Our hypothesis is that adversarial examples "live" closer to the boundary of the class-optimal manifold (defined in App. A) due to their origin in images of a different class, and therefore have weaker connectivity to real examples of the predicted class.

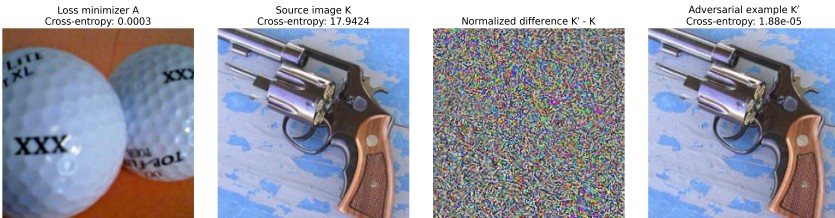

Figure 4: Adversarial attack. From the left: **i)** real input A that minimizes the loss for class *golf ball*, **ii)** source input K from a different class, **iii)** optimized pattern added to the K, and **iv)** adversarial example K' that minimizes the loss for the same class as A. The network predicts K' as a golf ball.

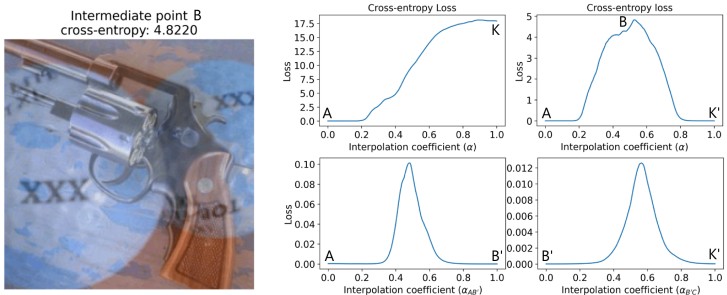

Figure 5: Barriers and path complexity. **Left:** Maximal loss point B on the primary barrier (between the real mode A and adversarial attack K'). **Right:** Interpolated paths between points from Figure 4. The primary barrier was bypassed through the optimized point B'. Secondary barriers (with lower loss) emerged on the new path A→B'→K' and can be further bypassed by a repeated application of the procedure to both segments of the new path (omitted here). This suggests that the path between a real mode A and an adversarial attack K' is more complex than that between two real images.

In addition to the complexity of the paths, we observed significant differences in the barrier heights and shapes between two key scenarios: barriers between real–real modes and real–adversarial modes. To assess the statistical significance of this effect, we conducted the following experiment. Seven unique pairs of images per class were selected from the validation subset of ImageNet, encompassing 1,000 classes in total. Two pairs from each class, having the highest loss difference between the inputs, were excluded. The remaining 5,000 pairs were interpolated with 250 steps in between, and loss curves were generated. In the adversarial branch, pairs were formed by combining one example from the selected class with one from a randomly chosen different class. Adversarial examples were created in the same way as described above with the following settings: learning rate 0.005, $\lambda_{\mathrm{MSE}} = 0.1$, $\lambda_{\mathrm{hf}} = 1e - 8$, and 512 iterations. The subsequent procedure mirrored the one described earlier, yielding another set of 5,000 filtered pairs (5 per class). Descriptive statistics for the maximum barrier height and the gap (the difference between the peak height and the loss value of the higher mode/image) are depicted in Figure 6.

Real–adversarial mode pairs exhibit significantly higher barrier max, with mean and median 5.82 and 5.99, respectively. In contrast, real–real pairs have mean 2.27 and median 1.79. The effect is even more pronounced for barrier gaps, where real–adversarial have mean 5.19 and median 5.30, but real–real have mean 1.00 and median only 0.47. The described behavior was utilized to design a new method for adversarial detection, which relies solely on the shape of the loss curves on linear interpolant paths.

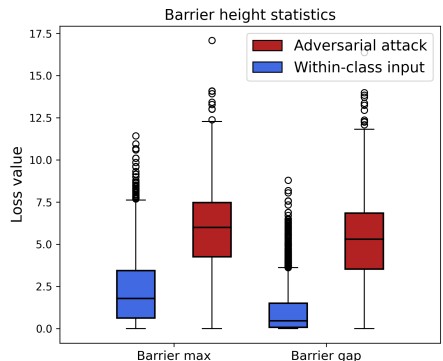

Figure 6: Barrier height statistics. Boxplots depict the maximum barrier heights and gaps (i.e., the difference between the barrier height and loss value of the higher-loss mode) for selected pairs from the ImageNet validation dataset. The barriers for real–real and real–adversarial example pairs are compared. A total of 5,000 representative pairs (5 per class) were utilized for each scenario.

#### 4.2.1 ADVERSARIAL DETECTION

Enhancing the adversarial robustness of models is essential across various domains, particularly in safety-critical applications (Pang et al., 2018). However, the majority of defenses (e.g., adversarial training) are not sufficient against advanced attacks and can be easily fooled by adaptive attacks (Tramer et al., 2020). Instead of maintaining model accuracy against adversarial examples, an alternative is to detect attacks and invalidate predictions for such inputs (Han et al., 2023; Harder et al., 2021).

We propose a simple algorithm for adversarial detection that leverages the relative lack of linear connectivity in natural–adversarial image pairs (as shown in Figure 6); First, we compute the (linear-path) loss curves for all training images $x_i$. Each loss curve is obtained from an input pair consisting of $x_i$ (a training image) and $x_{\text{template},y_i}$ (a low-loss reference image from the same class, $y_i = \text{argmax } f(x_i, \theta)$), which is preselected for each class. Second, concatenate the loss curves with logits $f(x_i, \theta)$, sorted in descending order, which provide information about prediction confidence. The second step slightly enhances performance against simpler attacks, though it can be fully ablated for attacks like deepfool and C&W). Third, resulting feature vectors (loss curves+logits) are then used to train a classifier. Test images are processed through the same pipeline and the resulting loss curves are classified by the model.

We implemented the algorithm to an existing benchmark (Harder et al., 2021) that uses VGG-16 (Simonyan & Zisserman, 2015) model and CIFAR10/100 datasets. The benchmark employs a representative selection of adversarial attacks (FGSM (Goodfellow et al., 2015), BIM (Kurakin et al., 2017), PGD (Madry et al., 2019), deepfool (Moosavi-Dezfooli et al., 2016), and C&W (Carlini & Wagner, 2017)) all in their untargeted forms. Interestingly, our method outperformed baselines for more advanced attacks (deepfool and the particularly challenging C&W) but underperformed on simpler (FGSM, BIM, PGD). This is a consequence of the nature of untargeted attacks, which generally aim to alter the prediction by maximizing the loss for the correct class. However, unlike FGSM, BIM, and PGD, both deepfool and Carlini & Wagner (C&W) also minimize the loss for a specific incorrect class, making the misclassification more controlled. Our algorithm relies on the existence of a distinct loss barrier, which is less pronounced in simpler attacks but more defined in advanced ones like deepfool and C&W. Complete results are provided in Table 1. Note that we used a classifier (KNN) that was optimized jointly for all attacks on validation data (in contrast to attack-dependent classifiers and hyperparameters in the benchmark). Additionally, unlike the best-performing baseline, our method does not require access to the feature maps.

Note that the proposed detection method could fail against adaptive attacks, as is common for the majority of current defenses (Bryniarski et al., 2021). We did not test this scenario due to the non-differentiability of our current version of the detection algorithm and will address it in future work.

### 4.3 THE EXISTENCE OF CONNECTIVITY IN UNTRAINED MODELS

Finally, we show that modes, represented by class-optimal inputs, are connected even in the case of untrained, randomly initialized models. This experiment was inspired by findings in Section 5, but we present it here, preceding the section, to maintain the consistency of the manuscript.

Table 1: (adapted from (Harder et al., 2021)) Comparison of detection methods. Accuracy/AUC (%). The attacks are applied on the CIFAR-10/100 test set and the VGG-16 NET. Attacks are described in Appendix D.

| Dataset | Detector | FGSM | BIM | PGD | Deepfool | C&W |
|---------|----------|------|-----|-----|----------|-----|
| CIFAR-10 | LID | 86.4 / 90.8 | 85.6 / 93.3 | 80.4 / 90.0 | 78.9 / 86.6 | 78.1 / 85.3 |
| | Mahalanobis | 95.6 / 98.8 | 97.3 / 99.3 | 96.0 / 98.6 | 76.1 / 84.6 | 76.9 / 84.6 |
| | InputMFS | 98.1 / 99.7 | 93.5 / 97.8 | 93.6 / 97.9 | 58.0 / 60.6 | 54.7 / 56.1 |
| | LayerMFS | **99.6 / 100** | **99.2 / 100** | **98.3 / 99.9** | 72.0 / 80.3 | 69.9 / 77.7 |
| | LayerPFS | 97.0 / 99.9 | 98.0 / 99.9 | 96.9 / 99.6 | 86.1 / 92.2 | 86.8 / 93.3 |
| | Mode connectivity (ours) | 69.8 / 76.8 | 95.2 / 98.6 | 91.8 / 97.4 | **91.4 / 96.9** | **93.7 / 98.3** |
| CIFAR-100 | LID | 72.9 / 81.1 | 76.5 / 85.0 | 79.0 / 86.9 | 58.9 / 64.4 | 61.8 / 67.2 |
| | Mahalanobis | 90.5 / 96.3 | 73.5 / 81.3 | 76.3 / 82.1 | **89.2 / 95.3** | 89.0 / 94.7 |
| | InputMFS | 98.4 / 95.5 | 89.1 / 94.1 | 90.9 / 95.1 | 58.8 / 62.2 | 53.3 / 54.6 |
| | LayerMFS | **99.5 / 100** | **97.1 / 99.5** | **97.0 / 99.7** | 83.8 / 91.0 | 87.1 / 93.0 |
| | LayerPFS | 96.9 / 99.3 | 90.3 / 96.7 | 92.6 / 97.6 | 78.8 / 84.4 | 79.1 / 84.0 |
| | Mode connectivity (ours) | 67.3 / 72.8 | 84.3 / 92.5 | 88.9 / 95.2 | 86.8 / 94.0 | **92.9 / 97.3** |

As there are no natural datasets for untrained models, we computed class-optimal synthetic inputs using the FVO (see Section 3.2 for definition, and Appendix A for analogical results from the FVO on a trained model). The optimization procedure failed to converge on GoogLeNet, likely due to the high input dimension (3, 224, 224). Using the ResNet18, adapted for CIFAR10 (input shape (3, 32, 32), output (10)), starting from Gaussian noise $\mathcal{N}(0, 0.01)$, we were able to find inputs with losses below the specified threshold 0.0005. Adam optimizer with learning rate 0.05 and weight decay 1e-7 was used for a maximum of 4096 iterations or until reaching the desired loss. A diversity of inputs was achieved through high-frequency penalization with a weight $\lambda_{\text{hf}} = 2.5e-7$ of one of the inputs in each pair.

The two modes were linearly interpolated, and the loss curve is shown in Fig. 7 left. After two rounds[1] of the standard mode-connecting procedure, we obtained a four-segment piece-wise linear path. $A \to B'_{S1} \to B' \to B'_{S2} \to C$ that has loss below $\delta = 0.001$ everywhere. Note that points $B'_{S1}$ and $B'_{S2}$ were obtained by optimizing the secondary barriers $B_{S1}$ and $B_{S2}$. All the loss curves and barriers are depicted in Fig. 7. In Appendix F.1, we also show how the linear connectivity (primary loss curve, i.e., loss on the linear interpolant path) changes during the training.

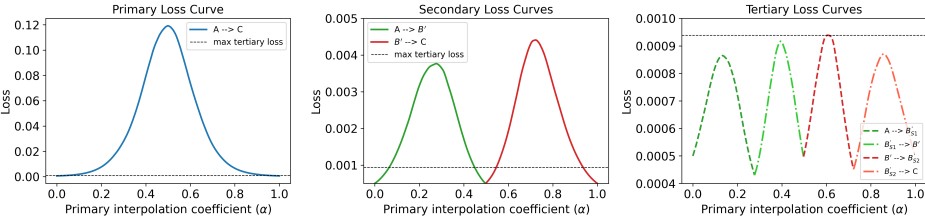

Figure 7: Input connectivity in untrained model. Loss curves for synthetic optimal inputs computed for an untrained model. **Left:** Primary loss curve, connecting the two optimal inputs. **Middle:** Secondary loss curves, connecting two original inputs through the intermediate point B′ (optimized from the primary barrier). **Right:** Tertiary loss curves, four-segment path connecting two original inputs through intermediate points $B'_{S1}$, B′, and $B'_{S2}$ (optimized first secondary barrier, optimized primary barrier, optimized second secondary barrier, respectively).

## 5 TOWARDS A THEORETICAL EXPLANATION OF INPUT SPACE CONNECTIVITY

This section establishes the theoretical foundations underlying the phenomenon of input space mode connectivity. We begin by introducing some definitions.

We denote our input and output spaces as $X = \mathbb{R}^{d_x}$ and $Y = B^{d_y} = \{y \in \mathbb{R}^{d_y} \mid \|y\| \le 1\} \subseteq \mathbb{R}^{d_y}$, respectively. A path connecting $x(0) \in X$ to $x(1) \in X$ is defined as a continuous function

---

[1]The first round optimizes the barrier on the primary linear path, the second round optimizes secondary barriers on the two linear segments and results in a four-segment path.

$x : [0, 1] \rightarrow X$. We define a network that maps the input space to the output space as any function within the framework of the tensor program formalism Yang (2019; 2020); Yang & Littwin (2021), featuring Lipschitz continuity-bounded point-wise nonlinearities. Specifically, for a parameter vector $\theta$, such a network accepts inputs and returns outputs:

$$f(\cdot; \theta) : X \rightarrow Y. \tag{4}$$

Given any $\delta > 0$, a network $f$, a parameter vector $\theta$, and a loss function $\mathcal{L} : Y \times Y \rightarrow \mathbb{R}^+$, we say that a path $x(\cdot)$ is $\delta$-**connected around** $y \in Y$ for this loss function if and only if:

$$\forall \alpha \in [0, 1] : \mathcal{L}(f(x(\alpha); \theta), y) \leq \delta, \tag{5}$$

with arbitrarily large probability.

A detailed description of our network class, and the requirements from the loss function is provided in appendix E.1. Further assumptions and limitations are discussed in appendix H.1.

### 5.1 GEOMETRIC MODE CONNECTIVITY

We present the concept of **geometric mode connectivity**, suggesting that almost all inputs on which a neural network makes similar predictions tend to be connected, as $d_X$ grows to infinity. We find this is the case empirically for both trained and untrained networks. We believe the latter finding can be proven as a consequence of high-dimensional geometry, and formalize it in the conjecture:

**Conjecture 5.1** (Geometric Input Space Connectivity).
*Given a subset of the input space $X' \subseteq X$, a network $f(\cdot; \theta)$ at initialization, and a loss function $\mathcal{L}$, whose specifications are provided in Appendix E.1, the following holds: For arbitrarily small $0 < \delta' < \delta$, any two inputs $x_0, x_1 \in X'$, selected independently of $\theta$ and with similar predictions are almost always connected as $d_X \rightarrow \infty$:*

$$P\left(x_0, x_1 \text{ are } \delta\text{-connected} \mid \mathcal{L}\left(f\left(x_0; \theta\right), f\left(x_1; \theta\right)\right) \leq \delta'\right) = 1 - O\left(e^{-d_X \tilde{\delta}}\right), \tag{6}$$

*where $\tilde{\delta} = O(\delta)$, and $x_0, x_1, \theta$ are chosen randomly as described in appendix E.1.*

We justify the conjecture for $\mathcal{L}(y, y') = \|y - y'\|$, and the case where the network's outputs are numbers between zero and one, $f(\cdot; \theta) : X \rightarrow [0, 1]$. The generalization to more general cases is straightforward.

We include a detailed proof sketch for this conjecture in appendix E, which we believe provides significant insight into the phenomenon of mode connectivity but note that it is not a fully complete and rigorous proof:

**(i)** First, in appendix E.2, we demonstrate that the network is Lipschitz continuous, by leveraging the Lipschitz continuity of the network's activation functions, and its semi-linear structure. Using this property, we show that for every $0 < \delta$ (assuming for simplicity that $\frac{1}{\delta} \in \mathbb{N}$), there exists $0 < \varepsilon$, such that if we divide the input space into $\varepsilon$-sized hypercubes, then for every cube inside the lattice, all the points' outputs ($f(x; \theta)$) belong to one of the following overlapping intervals:

$$Y_\delta = \left\{[0, \delta], \left[\frac{\delta}{2}, \frac{3\delta}{2}\right], [\delta, 2\delta], \ldots, [1 - \delta, 1]\right\}. \tag{7}$$

We can then label each input by the interval (or intervals), that corresponds to the output of all the inputs in the cube. We show that if there exists a sufficiently small $\varepsilon$, then for any two points $x_0, x_1 \in X'$ that satisfy the conjecture's condition ($\mathcal{L}(f(x_0; \theta), f(x_1; \theta)) \leq \delta'$), their respective cubes share the same interval. This $\varepsilon$ is chosen as the length of our graph's hypercubes.

**(ii)** Next, we demonstrate in appendix E.3 that to find a $\delta$-connected path between the two inputs, it suffices to find a connected path of cubes between the cube of the first input and the cube of the second, with the same interval-label.

Assuming the cube's intervals are drawn randomly and independently from each other with similar probability $p \approx \delta$, we encounter a **classic high-dimensional percolation problem** Duminil-Copin (2017); Meester & Roy (1996), where the cubes represent the points in the graph, and they are

connected if they share the same interval. It is well known that in such a case, the probability that two points are connected grows rapidly with the dimension, as shown in the conjecture 5.1.

It is important to note that the assumption of independent probability is not realistic, as there are strong correlations among nearby inputs of wide neural networks, even at initialization. This is why we decided not to consider statement 5.1 as a theorem, but rather as a conjecture. However, we expect that in general, assuming independence only underestimates the true connectivity, as the correlation between outputs of nearby inputs in neural networks at initialization tends to be positive.

## 5.2 PATH SIMPLICITY BIAS

As discussed above, we justified conjecture 5.1 only for neural networks at initialization, which raised the question, whether this property persists after training. In practice, we observed that not only do the inputs tend to remain connected, but they also tend to be connected approximately linearly, with only a relatively small barrier. Furthermore, by combining only a few linear paths, we can almost completely eliminate this barrier. This tendency is strong enough to be useful in adversarial detection, as we shown in section 4.2. We speculate that this phenomenon results from *implicit regularization*.

Implicit regularization refers to an inherent tendency of a learning system to prefer simpler hypothesis functions Soudry et al. (2018); Neyshabur (2017); Barrett & Dherin (2020); Gidel et al. (2019); Belkin et al. (2018); Jacot et al. (2018); Lee et al. (2019); Mulayoff et al. (2021). We propose that one kind of simplicity preferred by deep learning is generally not to change between similar inputs if possible. This tendency, combined with geometric considerations, could explain the observed behavior. In appendix F, we present some ideas on the mechanisms responsible for this tendency; however, more work is needed to substantiate this.

## 6 CONCLUSION AND FUTURE WORK

We demonstrated mode connectivity in the input space of deep networks trained for image classification, extending the original concept beyond parameter space. This phenomenon was shown with both real and synthetic images in various controlled setups. For tested examples, we were always able to find low-loss paths between any two modes. We explored potential applications of this connectivity, particularly for the adversarial detection and interpretability of DNNs. Our findings reveal that natural inputs can be distinguished from adversarial attacks by the height of the loss barrier on the linear interpolant path between modes. While for real–real mode pairs, the loss barrier is small or even negligible, real–adversarial pairs often have high and complex barriers. By exploring connections between different types of class-optimal inputs we obtained a novel perspective on DNN interpretability. The new evidence for mode connectivity beyond the parameter space supports the hypothesis that mode connectivity is an intrinsic property of high-dimensional geometry and can be studied through percolation theory. This hypothesis, however, requires additional investigation to fully understand its implications.

Future work could formalize the theory behind our conjecture of geometric mode connectivity and identify minimal requirements for its manifestation. Another direction is to develop a theory and conduct experiments to better understand implicit regularization in the input space. Practical applications include optimizing the proposed naive adversarial detection algorithm and leveraging input space connectivity insights for more effective adversarial training, e.g., sampling adversarial examples along low-loss paths for model retraining, analogous to parameter space connectivity in deep ensembles (Fort et al., 2020). Investigating the geometric properties of the input space loss landscape and exploring the motivations ford may also enhance model interpretability.

## ACKNOWLEDGMENTS

J.V. acknowledges the support of the Technology Agency of the Czech Republic under project FW11020175. The work of Y.O. is supported in part by Israeli Science Foundation excellence center, the US-Israel Binational Science Foundation, and the Israel Ministry of Science.

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

## A    TOWARDS OPTIMAL INPUT MANIFOLD

Here, we apply input space mode connectivity for the interpretability of DNNs. Synthetic images offer a controlled environment for examining input space mode connectivity. Here, images were generated using the feature visualization by optimization technique on GoogLeNet trained on ImageNet. Starting from Gaussian noise $\mathcal{N}(0, 1)$, we optimized a surrogate objective function (dot product $\times$ sqrt of cosine similarity $\times$ $1/2$), instead of directly optimizing for cross-entropy. We used this heuristic approach to generate synthetic inputs that better match human visual perception, as cross-entropy optimization, for unclear reasons, is less effective for this purpose. Through varied regularizations and transformations during optimization, we obtained two distinct modes, each exhibiting virtually zero cross-entropy loss, as illustrated in Figure 8. Mode A exhibits partial resemblance to natural objects of its class (*golf ball*). Mode C contains only high-frequency patterns or noise, lacking obvious semantic structure, and was chosen intentionally for this purpose. The bottom row of Figure 8 demonstrates that even such different types of synthetic modes can be connected. Moreover, since the modes exhibited effectively zero loss, there are no secondary barriers after optimizing the maximal loss intermediate point B (almost up to the numerical precision).

It is evident that a continuum of optimal inputs exists for a given class within the model. By employing a systematic approach to find and connect these optimal modes, the class-optimal manifold can be partially explored and analyzed. Formally, we define the class-optimal manifold as $M_{c,\delta} = \{x \in \mathcal{X} \mid \mathcal{L}(f_\theta(x), y_c) \leq \delta\}$. This approach would provide a novel perspective on DNN interpretability, especially as it extends the concept of feature visualization by optimization (Olah et al., 2017), beyond discrete optimal inputs or activation atlases.

## B    FURHTER DETAILS ON THE OPTIMIZATION PROCEDURE

In this section, we define the regularization terms used in the optimization procedure described and used in Section 4. The optimization employs the Adam optimizer on the cross-entropy loss with

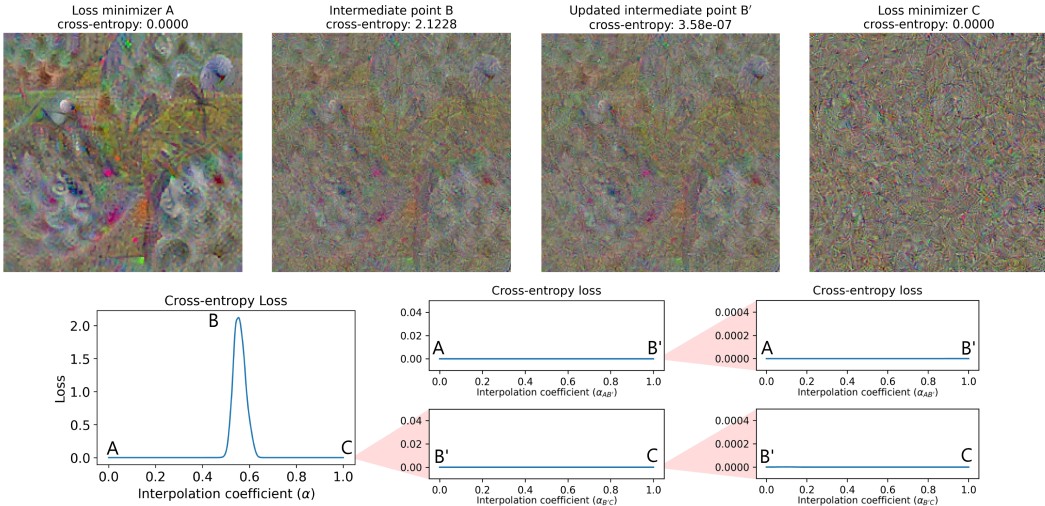

Figure 8: Synthetic modes. Synthetic images generated through input optimization from Gaussian noise. **Top row:** Modes A and C, interpolated high-loss point B and its optimized counterpart B'. **Bottom row:** The primary loss barrier was bypassed by a single round of optimization. The new path segments A→B' and B'→C are barrier-free up to several orders lower loss threshold.

two additional penalty terms: the deviation penalty $P_{\text{MSE}}$ and the high-frequency penalty $P_{\text{hf}}$, with corresponding strengths $\lambda_{\text{MSE}}$ and $\lambda_{\text{hf}}$, respectively.

The deviation penalty term is the Mean Squared Error (MSE) between the starting point (input tensor) and the resulting point (optimized tensor):

$$P_{\text{MSE}} = \frac{1}{d} \sum_{i=1}^{d} \left( x_i - x'_i \right)^2, \tag{8}$$

where $x_i$ represents the pixel value in the starting tensor, $x'_i$ is the corresponding pixel value in the optimized tensor, and $d$ is the total number of pixels in the tensor. This term controls the allowed deviation by adjusting its weight in the total loss.

The high-frequency penalty term is defined as:

$$P_{\text{hf}} = \sum_{i,j} |I(i, j + 1) - I(i, j)| + \sum_{i,j} |I(i + 1, j) - I(i, j)|, \tag{9}$$

where $I(i, j)$ represents the pixel value at position $(i, j)$ in the input tensor. The first term penalizes abrupt changes along the horizontal direction, while the second term does so along the vertical direction.

## C  PERCOLATION THEORY

Percolation theory, originally developed to describe the structure of a porous material, turned into a widely studied model in statistical physics and graph theory (Duminil-Copin, 2017).

Let $\mathcal{G}$ be an $N \times N$ grid, where at each cell, one inserts a ball with (Bernoulli distribution) probability $p$ and leaves it empty with probability $1 - p$. Site percolation theory considers the question: what is the probability that for any two nonempty cells $A$ and $B$ in $\mathcal{G}$, there is a path of nonempty cells that connects the two. When $p$ is small, the probability is zero, while when $p \to 1$, there is almost certainly a connecting path. The latter phase case is called percolation. In the limit $N \to \infty$ there is a sharp transition at a critical probability $p = p_c$, that separates the two phases. Figure 9 shows site percolation in two dimensions near the critical probability.

The percolation model has a straightforward generalization to higher dimensions, as well as to more than the two labels (empty and nonempty), and to a general probability distribution for choosing the

labels. In particular, the choice of labels between different cells can be correlated. The structure of the correlation affects the transition to percolation. Intuitively, one expects that higher correlations imply a higher probability of having connecting paths. Correlated percolation is an active field of research in physics and mathematics (see e.g. (Bug et al., 1985; Bricmont et al., 1987; Rodriguez & Sznitman, 2012; Chalhoub et al., 2024)).

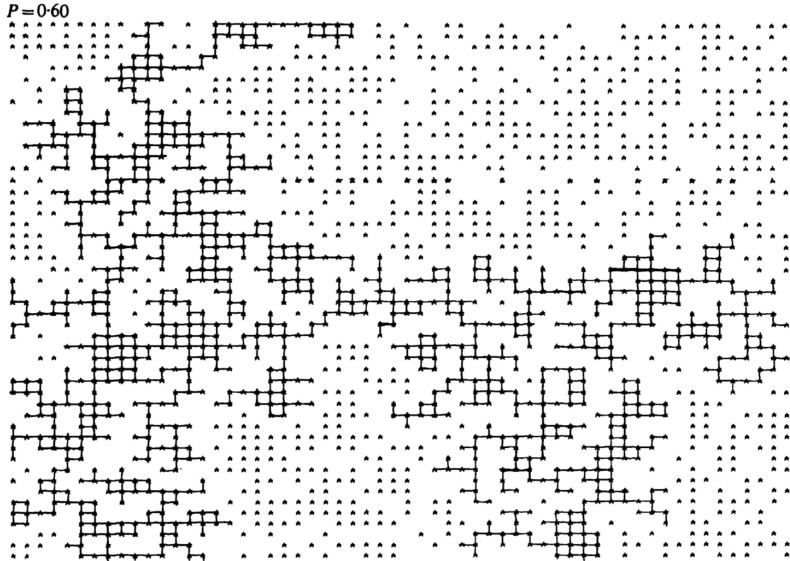

Figure 9: An illustration of site percolation in two dimensions near the critical probability Chalhoub et al. (2024) (the critical probability for two-dimensional site percolation is around 0.59). In this illustration, an "infinite" cluster can be observed. However, it is notable that a significant number of points do not belong to it.

In the framework of our work, the grid is defined by the $\varepsilon$-cells in the data space, while the labels are the intervals. Since the dimension of the grid is very large, the percolation probability $p_c$, which is inversely proportional to the dimension, is very small. Hence, our analysis at initialization is at the percolation phase.

We provide a dictionary (see Table 2) for the terms used in percolation theory in statistical physics and graph theory communities.

Table 2: Dictionary for percolation theory terms used in statistical physics and graph theory.

| statistical physics | graph theory |
|---|---|
| site | vertex (node) |
| bond | edge |
| cluster | connected component |
| infinite cluster | infinite connected component |
| percolation threshold | critical probability |
| order parameter | size of the largest component |
| lattice animal | connected subgraph (not infinite) |

## D  ADVERSARIAL DETECTION BENCHMARK

Here, we briefly overview attacks employed in the benchmark by Harder et al. (2021).

- **Fast Gradient Sign Method (FGSM)**: A rapid untargeted attack method that adjusts an input $x_i$ by adding a small perturbation in the direction of the loss function's gradient:

$$x_{i,\text{adv}} = x_i + \epsilon \cdot \text{sign}(\nabla_{x_i}\mathcal{L}(x_i, y)). \tag{10}$$

- **Basic Iterative Method (BIM)**: An iterative enhancement of FGSM, applying small, repeated perturbations, ensuring each update remains within an $\epsilon$-bounded range:

$$x_{i,\text{adv}}^{(n+1)} = \text{Clip}_{x_i,\epsilon}(x_{i,\text{adv}}^{(n)} + \epsilon \cdot \text{sign}(\nabla_{x_i}\mathcal{L}(x_{i,\text{adv}}^{(n)}, y))). \tag{11}$$

- **Projected Gradient Descent (PGD)**: A generalization of BIM that begins from a random point near $x_i$ within the $\epsilon$-ball.

- **Deepfool (DF)**: Iteratively modifies $x_i$ to cross the closest decision boundary by minimizing the perturbation required to alter the model's classification.

- **Carlini & Wagner (C&W)**: A powerful optimization-based attack that minimizes the $L_2$ norm of the perturbation while ensuring misclassification by optimizing a loss function designed to reduce the model's confidence in the correct class. The attack solves the following objective:

$$\min \left\| \frac{1}{2}(\tanh(x_{i,\text{adv}}) + 1) - x_i \right\|^2 + c \cdot f\left(\frac{1}{2}(\tanh(x_{i,\text{adv}}) + 1)\right), \tag{12}$$

  where $f(z)$ is a function that encourages the logits to either lower the true class's score (untargeted) or raise the target class's score (targeted). The parameter $c$ is optimized to balance minimizing the perturbation and ensuring successful misclassification. The C&W attack can be applied in both targeted and untargeted forms.

# E CONJECTURE 5.1 - PROOF SKETCH

## E.1 ASSUMPTIONS AND GENERALIZATIONS

### E.1.1 THE NETWORK

We justify conjecture 5.1 for fully connected neural networks defined by a set of $L \in \mathbb{N}$ layers, characterized by $L$ weight matrices and $L$ bias vectors, denoted as $\{\theta^{l,l-1}, \theta^l \mid l = 1, \ldots, L\}$. The first layer is defined as:

$$f^1 = \theta^{1,0}x + \theta^1. \tag{13}$$

Each subsequent layer $l = 2, \ldots, L$ satisfies:

$$f^l = \theta^{l,l-1}\phi(f^{l-1}) + \theta^l, \tag{14}$$

where $\phi$ is a Lipschitz-bounded activation function that acts on the vector in a pointwise manner.

We then define the network's prediction using a non-linear final activation $\varphi : \mathbb{R} \to [0, 1]$ such that:

$$f = \varphi(f^L). \tag{15}$$

The generalization to other networks within the tensor programs formalism Yang (2019; 2020); Yang & Littwin (2021) (and then using $\varphi$) is straightforward, as long as the nonlinearities are Lipschitz. The classes of networks described by this formalism is very diverse, and includes most of the relevant types of neural networks, such as fully connected neural networks, recurrent neural networks, long short-term memory units, gated recurrent units, convolutional neural networks, residual connections, batch normalization, graph neural networks, and attention mechanisms.

The reason that we can generalize our result to any network in this class is that in our justification, we only use the semi-linear structure of fully connected neural networks, and, by definition, any network in this class can be described as a composition of global linear operations and pointwise non-linear functions.

### E.1.2 INITIALIZATION AND INPUT DISTRIBUTION

As we will see, we can always divide the input space into sufficiently small hypercubes. Therefore, it is not crucial how we initialize our network, as long as it remains well-defined as $d_x \to \infty$.

The only requirement is that for every $0 < p < 1$ and $x \in X'$, the probability for having every label in $Y_\delta$ is at least $\tilde{\delta}$, where $\tilde{\delta} = O(\delta)$, except for a subset of intervals, whose combined probability is

less than $1 - p$. Therefore, $p$ is the probability for the entire conjecture to hold and can be arbitrarily close to 1.

We conjecture that this holds for most well-normalized initializations and networks, and any finite, arbitrarily large $0 < R$-radius balls:

$$X' = B_R^{d_x} = \left\{ x \in \mathbb{R}^{d_x} \mid \|x\| \leq R \right\} . \tag{16}$$

### E.1.3 THE LOSS FUNCTION

We work with $\mathcal{L}(y, y') = \|y - y'\|$. However, the generalisation to any loss function that satisfies $\mathcal{L} : Y \times Y \to \mathbb{R}^+$, such as it is Lipschitz continues in both arguments, and for every $y \in Y$:

$$\mathcal{L}(y, y) = 0 , \tag{17}$$

is straightforward.

### E.2 DIVIDING THE INPUT SPACE

Now that we have provided all of the necessary definitions, we can proceed to justify the conjecture. We will begin by demonstrating that small changes in the input space generate small changes in the network's output.

**Lemma E.1** (Lipschitz Continuity of the Network).

A neural network, as described above, is almost always Lipschitz continuous at initialization. This means that for any probability arbitrarily close to one, $0 < p < 1$, there exists some $0 < M$, such that for every $x, \Delta x \in X$:

$$\|f(x + \Delta x; \theta) - f(x; \theta)\| \leq M \|\Delta x\| . \tag{18}$$

An immediate consequence of this lemma is that for the situation described in conjecture 5.1, for any probability arbitrarily close to one, $0 < p < 1$, there exists some $0 < \varepsilon$ such that for every $x, \Delta x \in X$, if $\|\Delta x\| < \varepsilon$, then:

$$\|f(x + \Delta x; \theta) - f(x; \theta)\| \leq \delta - \delta' \leq \delta. \tag{19}$$

This result implies that we can divide our input space into $O(2N)$ classes, as in equation 7. This means that if we divide our input space into cubes of size $\frac{\varepsilon}{\sqrt{d_x}}$, not only does every input in each cube belong to the same class, but also that if two different inputs $x_0, x_1 \in X$ satisfy $\|f(x_0; \theta) - f(x_1; \theta)\| \leq \delta'$, then their cubes share the same interval.

*Proof of Lemma E.1.*

We prove the lemma by induction. We start with the induction base - the first layer. Since $\phi$ is Lipschitz continuous, we know that there exists some $0 < m$ such that for every $r, \Delta r \in \mathbb{R}$:

$$|\phi(r + \Delta r) - \phi(r)| \leq m |\Delta r| , \tag{20}$$

which means that for every $x, \Delta x \in \mathbb{R}^{d_x}$:

$$\|\phi(x + \Delta x) - \phi(x)\|^2 = \sum_{i=1}^{d_x} (\phi(x_i + \Delta x_i) - \phi(x_i))^2 \leq \sum_{i=1}^{d_x} (m \Delta x_i)^2 = m^2 \|\Delta x\|^2 . \tag{21}$$

Thus, for every $x, \Delta x$:

$$\|\phi(x + \Delta x) - \phi(x)\| \leq m \|\Delta x\| . \tag{22}$$

Multiplying by $\theta^{1,0}$ and adding $\theta^1$, we get:

$$f^1(x + \Delta x) - f^1(x) = \theta^{1,0} \phi(x + \Delta x) + \theta^1 - (\theta^{1,0} \phi(x) + \theta^1) = \theta^{1,0} (\phi(x + \Delta x) - \phi(x)). \tag{23}$$

Using the subordinate norm of the matrix, we find:

$$\|f^1(x + \Delta x) - f^1(x)\| \leq \|\theta^{1,0}\| \|\phi(x + \Delta x) - \phi(x)\| \leq \|\theta^{1,0}\| m \|\Delta x\| . \tag{24}$$

Which means that for every $0 < p < 1$, we can bound the change in $f^1(x + \Delta x) - f^1(x)$ by:

$$\|f^1(x + \Delta x) - f^1(x)\| \leq m' \|\Delta x\| . \tag{25}$$

We can continue this process by induction, showing that the same holds for every layer, which completes our proof. $\square$

**Remark E.1.**

It should be noted that the bound we found using the subordinate norm was sufficient for our needs, however, it is far from being the optimal one.

This is not a problem for our work, as all that we require is any Lipschitz bound, and then we can take $\varepsilon$ to be arbitrarily small. However, if one wishes to find a better bound, which also depends on the distance between inputs, or to investigate neural networks in the infinite limit appropriately, more care will be needed.

### E.3 FINDING THE PATH

To find now a $\delta$-connected path between the two inputs, all we need to do is to find a connected path of cubes between the cube of the first input to the cube of the second with the same intervals.

Assuming that the cube's intervals are drawn randomly and independently from each other with similar probability $p = O(\delta)$, we encounter a **classic high-dimensional percolation problem**, where the cubes represent the points in the graph, and they are connected if they share the same interval. It is well known that in such cases, the probability that two points are connected grows rapidly with the dimension, as shown in the conjecture. Specifically, if we ask what is the probability that a certain point will be part of the infinite cluster, we know that in the high-dimensional limit, we can neglect closed loops Stauffer & Aharony (2018). This is known as a "mean field approximation."

The probability that one input is not part of the infinite cluster is the probability that all nearby cubes connected to it are also not part of the infinite cluster. Thus, the probability that a point is part of the infinite cluster $P$ satisfies:

$$1 - P = (1 - pP)^{d_x} \lesssim e^{-d_x pP} \to P \gtrsim 1 - e^{-d_x pP} . \tag{26}$$

Defining $q = d_x p$, we find a lower bound for $P$ by solving the equation:

$$P = 1 - e^{-qP}. \tag{27}$$

Defining $Q = 1 - P$, we get:

$$1 - Q = 1 - e^{-q(1-Q)} = 1 - e^{-q}e^{qQ} \approx 1 - e^{-q}(1 - qQ + O(q^2Q^2)) \to$$
$$Q = e^{-q}(1 - qQ + O(q^2Q^2)) \to (1 + qe^q)Q = e^{-q} + O(q^2Q^2). \tag{28}$$

Which implies:

$$Q = \frac{e^{-q}}{1 + qe^q} + O(q^2Q^2) = O(e^{-q}) = O(e^{-pd_x}), \tag{29}$$

thus completing our justification.

## F (APPROXIMATE) LINEAR CONNECTIVITY

The next phenomenon we wish to address is the tendency of trained networks to exhibit linear paths. Intuitively, this is not particularly surprising. For any system to generalize effectively, it must exhibit a preference for simpler hypothesis functions. In the context of overparameterized models, this preference is often referred to as *regularization*, and when not explicitly imposed, it is termed "implicit regularization". A reasonable form of simplicity involves not varying drastically between similar inputs.

Extensive research has been conducted on implicit regularization from the perspective of parameter space. However, its understanding from the input space perspective remains limited. A notable study Mulayoff et al. (2021) demonstrated that at equilibrium, the second derivative of neural networks with respect to the inputs is bounded and can be minimized as the learning rate increases.

Another possible approach is to consider the Neural Tangent Kernel (NTK) limit. It is known that, in the infinite-width limit, neural networks exhibit linear-like behavior Jacot et al. (2018); Lee et al. (2019). In this limit, their evolution over time can be described by a kernel, a two-point matrix function $\Theta(x, x')$, which generally tends to be larger where the two inputs $x \simeq x'$ are similar.

In this regime, for gradient descent, the final prediction can be viewed as an "averaged sum" Lee et al. (2019) of the data, where for every $x \in X$, more weight is assigned to label of inputs that were

closer to $x$ (according to the kernel). And as it is reasonable to assume that a linear combination of two inputs will be closer to the original inputs than to any other input, that could partially explain the linear mode connectivity.

Another avenue that could be promising is to consider $f(\alpha x_0 + (1-\alpha)x_1, \theta)$, and decompose it as $f(\alpha x_0, \theta) + f((1-\alpha)x_1, \theta)$ plus an additional term, whose magnitude should be shown to be small. If both $f(\alpha x_0, \theta)$ and $f((1-\alpha)x_1, \theta)$ align in the same direction, we would obtain the correct label after applying the softmax function.

It is important to emphasize that the ideas presented here are speculative, and only represent our preliminary conjectures. We have not yet validated these concepts, and more rigorous research is needed to confirm or refute these notions.

### F.1 Connectivity evolution throughout training

Following the observation of connectivity in untrained models (Sec. 4.3) and hypothesizing the role of implicit regularization in trained models, we studied the temporal evolution of the connectivity over training batches and epochs. To maintain comparability for the early stages of training, where the model performs poorly on real data, we again employed synthetic optimal inputs. The optimization process was adapted from Sec. 4.3 (ResNet18, input shape $(3, 32, 32)$, output shape 10, starting from Gaussian noise $\mathcal{N}(0, 0.01)$, Adam optimizer with learning rate 0.05 and weight decay 1e-7, maximum of 4096 iterations or until reaching the desired loss), with difference of using the loss threshold $\delta = 0.005$ for convenience and faster convergence. A diversity of inputs was achieved through high-frequency penalization with weight $\lambda_{\text{hf}} = 2.5e - 7$ of one of the inputs in each pair. We prepared 5 pairs for each class and computed primary loss curves. Averaged loss curves (over all pairs, classes and model stages) for the first 30 batches are shown in Fig. 10, and similarly for the first 30 epochs in Fig. 11.

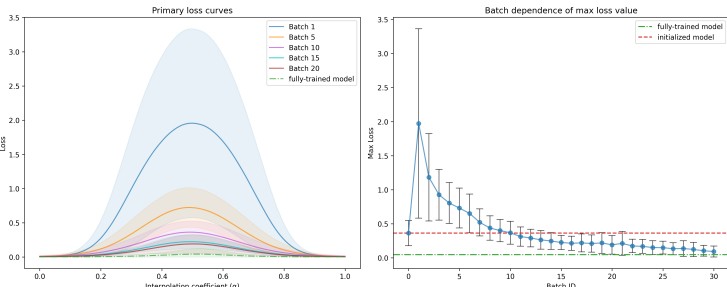

Figure 10: Batch evolution of the averaged loss barriers. Batches are indexed starting from one; zero means an untrained model. Shaded area and error bars show standard deviation.

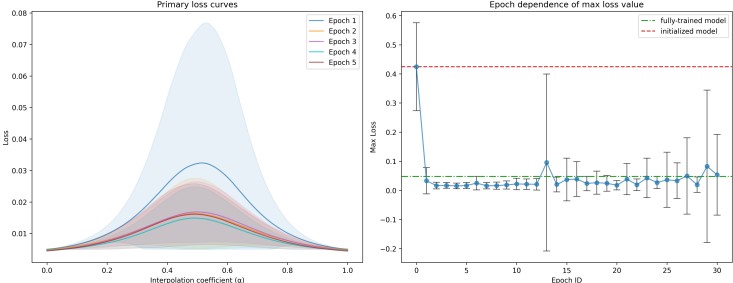

Figure 11: Epoch evolution of the averaged loss barriers Epochs are indexed starting from one; zero means untrained model. Shaded area and error bars show standard deviation.

While we observe the increase of connectivity (decrease of loss barriers) in the early phases of training, it saturates and variance begins to grow at some point. There are likely more competing effects taking part over the training. It is important to note that the barrier height and complexity depend on the losses of the interpolated inputs.

## G EXTRA RESULTS FOR THE SECTION 4.1

Here, we present additional examples of the input space mode connectivity on pairs drawn from the validation datasets of ImageNet and CIFAR10. We used the following models: GoogLeNet (see Figure 12), Vision Transformer (Figure 13), ResNet18 (Figure 14), MLP (Figure 15), and CNN (Figure 16).

The exact architectures that were employed are the following:

1. **GoogLeNet** - pretrained model `inception v1`.

2. **ViT** - `vit_base_patch16_224.sam_in1k` from `timm` library pretrained on ImageNet-1k using Sharpness Aware Minimization Chen et al. (2022) .

3. **ResNet18** - custom trained on CIFAR10.

4. **MLP** - custom trained on CIFAR10. The architecture consisted of 5-layer fully-connected layers with input dimension 3072 (32x32x3 images), hidden layers of sizes 4096, 4096, 2048, 1024, and output dimension 10. We used ReLU activations and dropout (p=0.1) after each hidden layer, trained for 100 epochs with Adam optimizer (lr=0.001, weight_decay=1e-6).

5. **MLP** - custom trained on CIFAR10. The architecture consisted of six layers (4 convolutional, 2 fully-connected), with convolutional layers of sizes 64, 128, 256, 512 (3x3 kernels), followed by fully-connected layers of sizes 1024 and 10. We used batch normalization, ReLU activations, max pooling (2, 2), and dropout (p=0.1) after each layer, trained for 100 epochs with Adam optimizer (lr=0.001, weight_decay=1e-6).

Note that the loss barrier can be negligibly small for some pairs even before the initial round of barrier optimization.

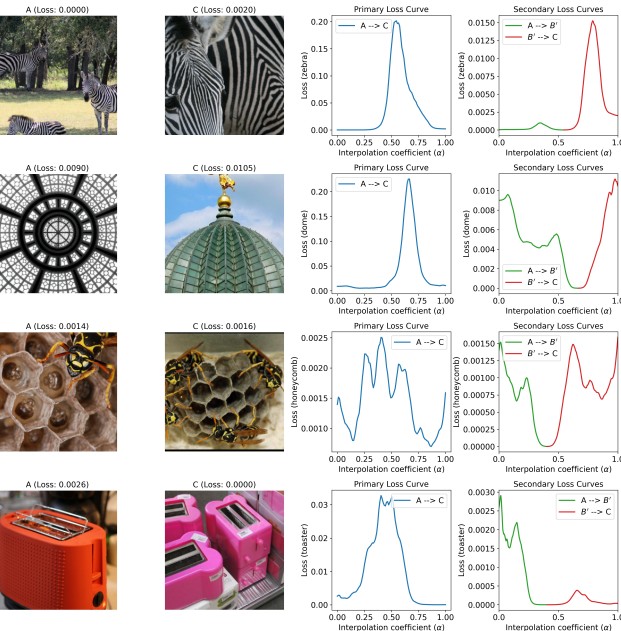

Figure 12: GoogLeNet(ImageNet) loss curves. Representative examples of connectivity in a trained GoogLeNet for selected classes. Each row contains a pair of same-class images. The primary loss curve is obtained from linear interpolation A→B, and a secondary loss curve A→B'→C. Note that the highest loss value in the secondary curves is at least one magnitude lower than in the primary curves.

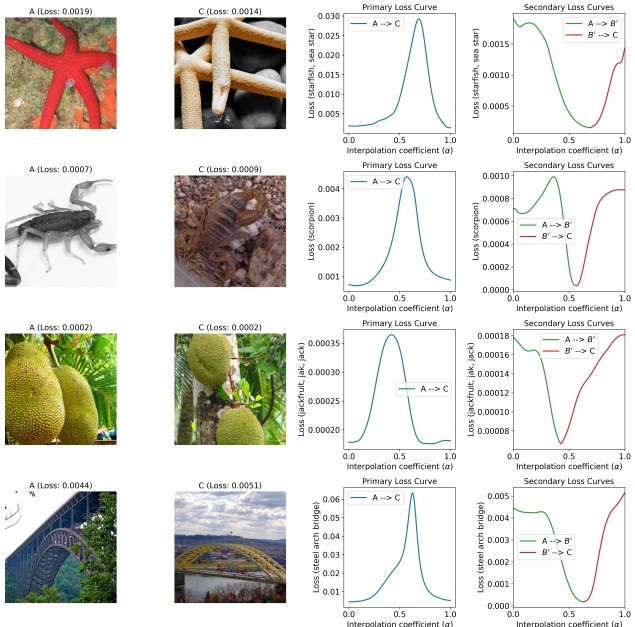

Figure 13: Vision Transformer(ImageNet) loss curves. Representative examples of connectivity in a trained ViT for selected classes. Each row contains a pair of same-class images. The primary loss curve is obtained from linear interpolation A→B, and a secondary loss curve A→B'→C. Note that the highest loss value in the secondary curves is at least one magnitude lower than in the primary curves.

## G.1 Commentary on the ViT results

The results presented in Figure 13 are valid for standard transformer models without advanced regularization strategies or other training tricks. For more recent models employing data augmentations, regularization, and extensive pretraining (such as Steiner et al. (2022)), the lowest observed loss for validation images was approximately 0.05. These loss values exceed the threshold for our definition of a mode, and consequently, their connectivity is qualitatively different. In our non-extensive experiments, we were able to connect these higher-loss examples as well, but the number of required segments varied, typically three or more.

## G.2 Small barriers close to numerical precision

In some cases, for the custom-trained MLP and CNN models, the loss barriers are extremely small. After a single round of our optimization procedure, the loss values approach zero within numerical precision (see Figure 17).

Specific cases exhibit "true" linear connectivity (see Figure 18), which is not generally observed in advanced models (beyond the MLP and CNN architectures). Our hypothesis is that this behavior results from train-test data leakage in CIFAR10 and the memorization of these data samples. This phenomenon will be studied more extensively in future work.

## H Limitations

### H.1 Section 5.1

The main limitation, other than the basic assumptions discussed in section E.1, is that we treated our problem as a standard model of percolation, assuming no correlation between different sites. However, this assumption does not hold for neural networks, not even at initialization. We argue,

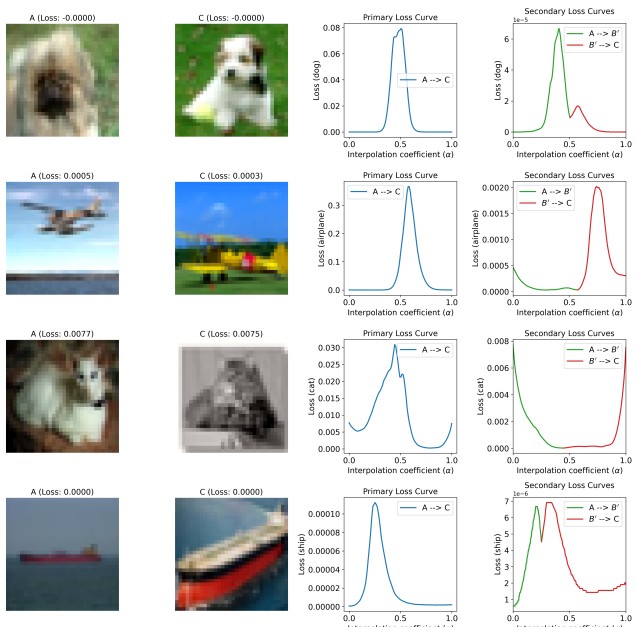

Figure 14: ResNet18(CIFAR10) loss curves. Representative examples of connectivity in a trained ResNet18 for selected classes. Each row contains a pair of same-class images. The primary loss curve is obtained from linear interpolation A→B, and a secondary loss curve A→B'→C. Note that the highest loss value in the secondary curves is at least one magnitude lower than in the primary curves.

however, that since correlations in neural networks are positive, the actual scenario should be more favorable. Nevertheless, diligent investigation and exact proofs are necessary to confirm this.

## H.2 SECTION 5.2

The ideas discussed in that section are speculative and represent our initial hypotheses. These concepts have not yet been validated, and further rigorous research is required to either confirm or disprove them.

## H.3 SECTION 4

Experiments were carried out on common vision models. The potential generalization of the concept to other data modalities was not tested. Our approach requires a continuous input space up to standard numerical precision. Therefore, technical adjustments would be necessary for applications beyond vision models, such as language models that use tokenization.

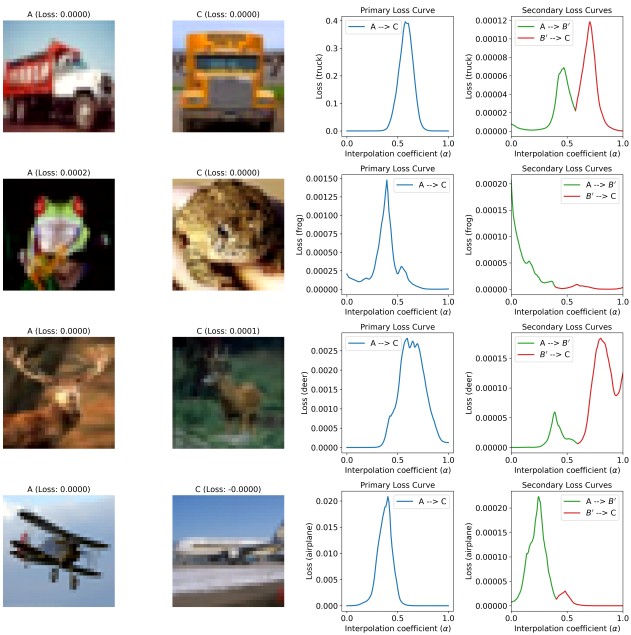

Figure 15: MLP(CIFAR10) loss curves. Representative examples of connectivity in a custom-trained MLP for selected classes. Each row contains a pair of same-class images. The primary loss curve is obtained from linear interpolation A→B, and a secondary loss curve A→B'→C. Note that the highest loss value in the secondary curves is at least one magnitude lower than in the primary curves.

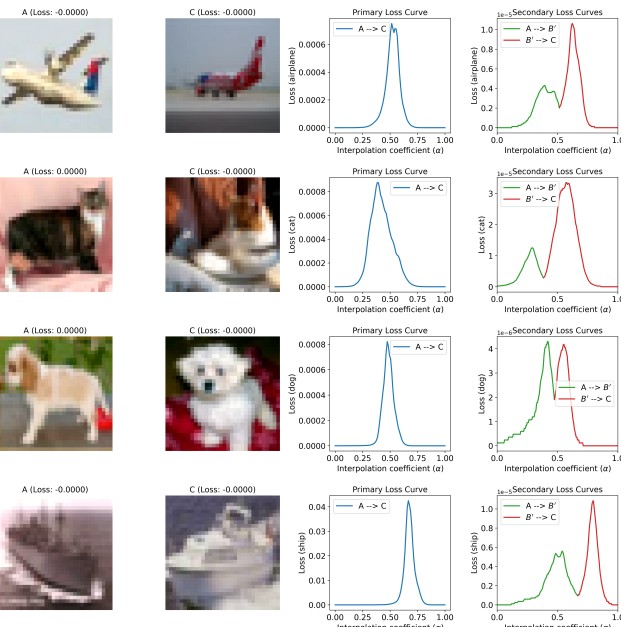

Figure 16: CNN(CIFAR10) loss curves. Representative examples of connectivity in a custom-trained CNN for selected classes. Each row contains a pair of same-class images. The primary loss curve is obtained from linear interpolation A→B, and a secondary loss curve A→B'→C. Note that the highest loss value in the secondary curves is at least one magnitude lower than in the primary curves.

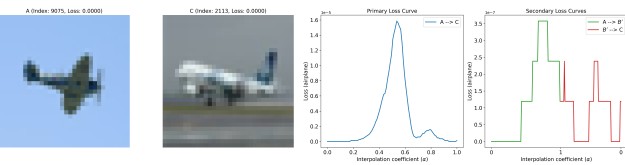

Figure 17: MLP (CIFAR10) loss curve example with a small barrier. Many example pairs have loss barriers so small that, after a single round of our optimization procedure, the loss values approach zero within numerical precision.

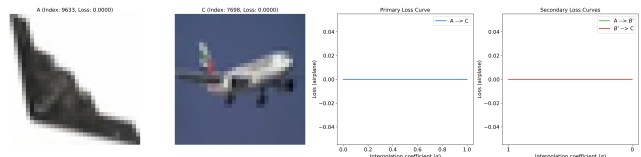

Figure 18: MLP (CIFAR10) loss curve example with zero barrier. Some example pairs have loss barriers exactly equal to zero and, therefore, exhibit true linear connectivity.

