# OpenReview forum: "Input Space Mode Connectivity in Deep Neural Networks"
_ICLR.cc/2025/Conference — ICLR 2025 Poster_

### Official Review · Reviewer_rBxt · 2024-11-03

**Soundness:** 3
**Presentation:** 2
**Contribution:** 3
**Rating:** 5
**Confidence:** 3

**Summary:**

This paper presents an interesting phenomenon, that given two input points $A,C$ which are classified as the same class by the model (i.e. have similar output), there exists a point $B'$, such that the linear interpolation between $A,B'$ and $B',C$ are all classifed as the same class. The authors refer to this phenomenon as "input space mode connectivity", despite they are not actually linear connected. The authors also presented a analysis under a very strong and unrealistic condition, conjecturing that this phenomenon is intrinsic to high-dimensional spaces.

**Strengths:**

- This phenomenon is interesting and potentially valuable for understanding the behaviour of deep neural networks and the geometry of high-dimensional spaces.
- This finding has a practical useness that it can be used to detect adversarial attacks.

**Weaknesses:**

- The presentation is kind of confusing. See Questions for concrete issues.
- In my understanding, the finding in this paper is not really linear connectivity as the path found by the proposed method is actually a piecewise lienear path with 2 pieces. This makes the title and introduction kind of misleading.
- The experiments are only on a few image classification tasks and models. It is not clear if this phenomenon is general enough.
- It seems that the thoretical explanation presented in Section 5.2 (even if we omit the too strong assumption of randomized labeling of each grid) only explains why there can exist a connected path, but does not guarantee that the path is linear, which is inconsistent with Conjecture 5.1.
- In Conjecture 5.1, why do you need to assume "a subset of input space $X' \subseteq X$"? Does the conjecture hold even if $X'$ itself is unconnected? Moreover, there is a statement "two random inputs $x_0, x_1 \in X'$". What is the distribution of the randomness? If it is uniform, then there must be extra constraints on $X'$ (such as compactness), since not every subset of a Euclidean space has a uniform distribution. (For example, you can not draw two points uniformly from a 2d plane).

**Questions:**

- I don't understand the algorithm presented in Section 4.2.1. In my understanding, the claim is that if one of the endpoints is from an adversarial attack, then the method of optimizing $B$ to find a connected (piecewise linear) path will not work. However, the algorithm of detecting adversarial attack is to use the interpolation loss curve and logits as the input and do a classification. I don't see how this proposed algorithm is related to the finding.
- The wording of Section 4.3 is too confusing that I totally can not understand. What does "natural datasets for untrained models" mean? What does "starting from Gaussian noise" mean (start from it and do what?) What does "high-frequency penalization" mean?
- In Conjecture 5.1, why does the condition has "for any probability $0 < p < 1$", but $p$ is never used in the statement? What does "almost always connected" mean?

---

### Official Review · Reviewer_hRew · 2024-11-04

**Soundness:** 3
**Presentation:** 2
**Contribution:** 2
**Rating:** 6
**Confidence:** 4

**Summary:**

This paper studies input space mode connectivity, where instead of studying the loss surface as a function of the parameters, the authors consider varying the inputs (for a fixed set of parameters). More specifically, the authors investigate paths connecting two sets of inputs and the resulting behaviour of the loss, in complete duality to the well-known mode connectivity in parameter space. Several choices for “modes” in this context are explored: (1) Validation data points that achieve very low loss, (2) a validation data point and an adversarial example optimized towards the same class but based on a datapoint of different class and (3) synthetic “optimal" points optimized to maximise a given logit. For all these cases, the authors show how simple piecewise linear paths suffice to connect such points, while limiting the barrier to very small values. Adversarial examples exhibit larger barriers in a significant manner, allowing a detector to leverage this difference to classify whether an image is adversarial or not.

**Strengths:**

1. The role of the input when it comes to loss behaviour is somewhat understudied and the authors develop new ideas in this direction while keeping things very analagous to the results observed for parameter loss landscapes.
2. The authors give further credibility to their results by mathematically proving them in an idealized setting assuming independence. While this is not realistic, I do find the argument of the authors convincing that correlations in this case will most likely help connectivity.

**Weaknesses:**

1. The biggest weakness is the lack of motivation presented in this paper for input space connectivity. Why is this an interesting quantity to study? In case of the parameter loss landscape where this notion originated, the motivation was from an optimisation point of view; is SGD attracted to a convex region of the parameter space? Does it find isolated minima or are there entire regions of low loss? ﻿There might be good motivations for input space connectivity as well (I’m not an expert in this area) but the paper does not do a good job at presenting it in its current form. In general I also have no intuition whether it is surprising that real-real images can be connected with two segments or not, etc.

2. I like the idea of using the difference in barrier between real-real and real-adversarial inputs, but as usual in adversarial robustness, I think that new threat models need to be investigated when taking this idea into account. I.e. can one now develop adversarial examples that are designed to mimick the barrier of real examples, thus fooling the new classifier? I don’t expect the authors to necessarily develop such an algorithm but this possibility should at least be discussed in the paper.

3. I also have a hard time interpreting the adversarial example results. It is not that surprising that it requires more segments to connect things properly compared to the real-real scenario (the image is still very different after all). How does this compare to simply two images coming from different classes and their barriers in between?

4. The writing of the paper is not very satisfying. The notation is rather sloppy (e.g. "0.1*MSE for image deviation and 1e-7*high-frequency penalty"),  optimization details are listed without defining them (e.g. high-frequency penalty). The actual algorithm to obtain a piece-wise linear curve is never properly defined.

**Questions:**

1. Do you have any intuition whether results would fundamentally change if other architectures were considered? E.g. [1] find that mode connectivity in the parameter-sense is influenced by the choice of architecture, i.e. there is different behaviour for vision transformers or multi-layer perceptrons.
2. I’m a bit confused regarding the adversarial example detector; Are you comparing loss barriers after a single iteration of your algorithm in both cases, or after two iterations in case of the adversarial setting? I thought that in both cases the barriers became very small?

[1] Disentangling Linear Mode-Connectivity, Altintas et al., 2023

---

### Official Review · Reviewer_KNNq · 2024-11-04

**Soundness:** 3
**Presentation:** 2
**Contribution:** 2
**Rating:** 6
**Confidence:** 3

**Summary:**

The authors identify an interesting phenomenon, namely input mode connectivity, where samples with similar predictions could be approximately linearly interpolated such that the interpolated sample remains a low loss.

**Strengths:**

1. This topic is interesting. Investigating the model connectivity in the input space could help us to shape the decision boundary of DNNs.
2. The insight of mode connectivity is indeed an intrinsic property of high-dimensional geometry is important, as it might be able to explain various phenomena lied in the field of model connectivity, such as wide neural networks are easier to satisfy mode connectivity after accounting for permutation invariance.
3. The potential application towards adversarial examples is insightful, which might explain the existence of adversarial examples.

**Weaknesses:**

1. The major issue of this work is that the investigation is not in-depth enough. For example, in Fig. 3, the path A->B'->C and A->B->C look similar but they differ significantly in terms of mode connectivity.
    - How should we quantify such differences? or why the small difference B'-B (as shown in right bottom of Fig. 1) is significant in terms of model connectivity?
    - Here is another example, in the adversarial example part, why real-adversarial pair shows a large barrier than real-real pair? An intuitive explanation is at least expected.

These are all important questions and represent the motivation why we are interested in investigating the mode connectivity in input space.

2. Their theory cannot explain the phenomenon they discovered. Their conjecture is only able to explain the mode connectivity for untrained NNs with infinity large input dimension. However, in their experiments, two real images with similar predictions are usually not connected unless another intermediate point is found, say B'. Clearly, their theory cannot explain the realistic scenarios.

**Questions:**

1. Can we repeat the produce of finding A->B'-C for each segment recursively to obtain A->E'->D'->B'->F'->G'-C (maybe a longer path), such that there is no essential barrier existing?
2. How can we use the input mode connectivity to give a picture of the decision boundary of DNNs?

---

### Meta-Review · Area_Chair_Ttmj · 2024-12-22

**Metareview:**

This paper identifies a new phenomenon, input space mode connectivity, that different images with similar predictions are generally connected by a simple path. The paper also presents some theoretical intuition suggesting that such a phenomenon could be generically true in high-dimensional spaces. The paper makes valuable contributions toward understanding the behavior of neural networks and the geometry of high-dimensional spaces. I believe this work is worth sharing with the community and recommend acceptance.

**Additional Comments On Reviewer Discussion:**

The authors satisfactorily addressed several confusion points initially raised by the reviewers. Additional experimental results for ViT, MLP, and simple CNN were added during the discussion phase as suggested by the reviewers. The AC does not see any major concerns remaining.

---

### Decision · Program_Chairs · 2025-01-22

Accept (Poster)